# Comparison of Ovarian Morphology and Follicular Disturbances between Two Inbred Strains of Cotton Rats (*Sigmodon hispidus*)

**DOI:** 10.3390/ani11061768

**Published:** 2021-06-12

**Authors:** Md Rashedul Islam, Osamu Ichii, Teppei Nakamura, Takao Irie, Akio Shinohara, Md Abdul Masum, Yuki Otani, Takashi Namba, Tsolmon Chuluunbaatar, Yaser Hosny Ali Elewa, Yasuhiro Kon

**Affiliations:** 1Laboratory of Anatomy, Department of Basic Veterinary Sciences, Faculty of Veterinary Medicine, Hokkaido University, Hokkaido 060-0818, Japan; Rashedul@vetmed.hokudai.ac.jp (M.R.I.); ichi-o@vetmed.hokudai.ac.jp (O.I.); nakamurate@vetmed.hokudai.ac.jp (T.N.); Masum@vetmed.hokudai.ac.jp (M.A.M.); yukiotani35@vetmed.hokudai.ac.jp (Y.O.); t-namba@vetmed.hokudai.ac.jp (T.N.); Tsolmon@vetmed.hokudai.ac.jp (T.C.); y-elewa@vetmed.hokudai.ac.jp (Y.H.A.E.); 2Department of Surgery and Theriogenology, Faculty of Animal Science and Veterinary Medicine, Sher-e-Bangla Agricultural University, Dhaka 1207, Bangladesh; 3Laboratory of Agrobiomedical Science, Faculty of Agriculture, Hokkaido University, |Hokkaido 060-0818, Japan; 4Department of Biological Safety Research, Chitose Laboratory, Japan Food Research Laboratories, Hokkaido 066-0052, Japan; 5Laboratory of Veterinary Parasitology, Faculty of Agriculture, University of Miyazaki, Miyazaki 889-2192, Japan; irie.takao.r0@cc.miyazaki-u.ac.jp; 6Medical Zoology Group, Department of Infectious Diseases, Hokkaido Institute of Public Health, Hokkaido 060-0818, Japan; 7Frontier Science Research Center, University of Miyazaki, Miyazaki 889-1692, Japan; mogura3@med.miyazaki-u.ac.jp; 8Department of Anatomy, Histology and Physiology, Faculty of Animal Science and Veterinary Medicine, Sher-e-Bangla Agricultural University, Dhaka 1207, Bangladesh; 9Department of Basic Science of Veterinary Medicine, School of Veterinary Medicine, Mongolian University of Life Science, Ulaanbaatar 17024, Mongolia; 10Department of Histology, Faculty of Veterinary Medicine, Zagazig University, Zagazig 44519, Egypt

**Keywords:** cotton rat, multi-oocyte follicles, double nucleated oocytes, proliferative cells, mitochondrial clouds

## Abstract

**Simple Summary:**

Multi-oocyte follicles have been reported in several mammals, especially in rabbits and hamsters, although their significance remains unclear. The present study compared ovarian histology, focusing on folliculogenesis, between two inbred cotton rat strains maintained at Hokkaido Institute of Public Health and the University of Miyazaki. Abundant multi-oocyte follicles and double-nucleated oocytes were observed in the Hokkaido strain, whereas Miyazaki had fewer multi-oocyte follicles and lacked double-nucleated oocytes. These findings indicate that early folliculogenesis events such as oocyte nest breakdown and oocyte vitality, rather than proliferation and cell death in each oocyte, affect the unique ovarian phenotypes found in cotton rats, including multi-oocyte follicles or double-nucleated oocytes, and their differences between strains. Therefore, these results can clarify mammalian folliculogenesis and its abnormal processes.

**Abstract:**

Most mammalian ovarian follicles contain only a single oocyte having a single nucleus. However, two or more oocytes and nuclei are observed within one follicle and one oocyte, respectively, in several species, including cotton rat (CR, *Sigmodon hispidus*). The present study compared ovarian histology, focusing on folliculogenesis, between two inbred CR strains, HIS/Hiph and HIS/Mz. At 4 weeks of age, ovarian sections from both the strains were analyzed histologically. Multi-oocyte follicles (MOFs) and double-nucleated oocytes (DNOs) were observed in all stages of developing follicles in HIS/Hiph, whereas HIS/Mz had MOFs up to secondary stages and lacked DNOs. The estimated total follicles in HIS/Mz were almost half that of HIS/Hiph, but interstitial cells were well developed in HIS/Mz. Furthermore, immunostaining revealed no clear strain differences in the appearance of oocytes positive for Ki67, PCNA, and p63 in MOF or DNOs; no cell death was observed in these oocytes. Ultrastructural analysis revealed more abundant mitochondrial clouds in oocytes of HIS/Hiph than HIS/Mz. Thus, we clarified the strain differences in the CR ovary. These findings indicate that early events during folliculogenesis affect the unique ovarian phenotypes found in CRs, including MOFs or DNOs, and their strain differences.

## 1. Introduction

In the mammalian ovary, the reserve of primordial follicles (PrF) serves as the source of developing follicles throughout the reproductive lifespan. The growing follicles that develop from this reserve constitute early in life, and the reserve of the PrF gradually decreases with both follicle growth activation and follicle degeneration during normal folliculogenesis. The process of folliculogenesis is similar in almost all mammals, but the timing of the formation of the reserve is species-specific [1]. Although folliculogenesis is a key event in reproductive physiology, the mechanisms involved in processes such as the appearance of multi-oocyte follicles (MOFs) [2,3,4] and double-nucleated oocytes (DNOs) [3,4], which are structures with two or more oocytes within a single oocyte follicle (SOF) and two nuclei within a single oocyte, respectively, still remain unclear. Most follicles contain only one oocyte, but the presence of MOFs has been described in several animal species, showing large species-related frequency differences, such as in cattle [5,6], dogs [7,8], pigs [9,10], and sheep [11,12]. Several researchers have hypothesized that MOFs are formed during the early stages of folliculogenesis. MOFs have been developed from natural polymorphisms that result in oocyte rearrangements [6,10], failure to separate the germ cells during the early phases of folliculogenesis [13], or the inclusion of various germ cells within a follicle [14]. In contrast, although DNOs are occasionally observed in humans with infertility [15], the species differences and formation mechanisms of DNOs remain unclear.

Cotton rats (CRs) are rodents belonging to the family Cricetidae (which includes hamsters and gerbils) and are distributed over most of the southern United States. There are seven species of CRs; of these, only *Sigmodon hispidus* (SH) and *Sigmodon fulviventer* are used as laboratory animals, and SH is more commonly used than *Sigmodon fulviventer*. SH is by far the most widely distributed CR species, ranging across the southern United States, from Virginia to southeastern California, southward through Mexico and Central America, into the tropical regions of northern America [16]. There are 25 subspecies of SH. These subspecies are well characterized and are commonly used in biomedical research [17]. In CRs, the first estrus cycle starts between 20 and 30 days of age (spring onset is at 30–40 days of age), the estrus cycle averages 9 days (4–20 days; not stable), gestation length is 27 days, and hair growth is present at birth [17]. Laboratory-reared SH females also undergo first estrus between 20 and 30 days of age and generally attain puberty between 30 and 40 days of age. Although SH is prolific in nature and produces nine litters per year, averaging five to six pups per litter, the reproductive characteristics of this species are not well known [17]. Furthermore, inbred SH develops unique phenotypes such as pharyngeal pouch remnants, metabolic disorders, carcinomas, anemia with renal inflammation, and chronic kidney diseases [18,19,20,21].

Rodents, especially laboratory mice belonging to the family Muridae, serve as an animal model for the study of folliculogenesis, but there have been no reports on the abundance of MOFs and DNOs. However, some researchers have found MOFs in rodents treated with synthetic estrogenic hormones [22] or the estrogen receptor β agonist 2,3-bis(4-hydroxyphenyl)-propionitrile [23], as well as in the ovaries of AQP8 knockout [24] or p27Kip1 mutant mice [25]. We have previously reported several follicular disturbances, including the abundance of MOFs and DNOs in an inbred strain of SH, HIS/Hiph, maintained at the Hokkaido Institute of Public Health [3,4]. It is still unknown whether these phenotypes are preserved in other SH strains or other CRs, and a comparative study would help to elucidate whether HIS/Hiph is an appropriate animal model to study folliculogenesis and the molecular mechanisms of these disturbances. Furthermore, the abundance of CR MOFs and DNOs could be important in future research focusing on the ovulation- and fertility-related features of oocytes derived from MOFs and DNOs to clarify the characteristics of their reproductive function in other domestic mammals and humans. In this study, we compared the histological and ultrastructural features of the ovary in HIS/Hiph with those in HIS/Mz, another inbred strain of SH.

## 2. Materials and Methods

### 2.1. Animals and Ovary Processing

All animal and experimental procedures followed the ethical principles and guidelines outlined by the Hokkaido Institute of Public Health (Sapporo, Japan; approval no. K27-03), and the University of Miyazaki (Miyazaki, Japan; approval no. 2020-503). Inbred strains of SH, i.e., HIS/Hiph and HIS/Mz, maintained at the Hokkaido Institute of Public Health and University of Miyazaki, respectively, were used in the present study. All SHs were maintained in a conventional room, and food and water were provided *ad libitum*. They were euthanized by cutting the abdominal aorta while deeply anesthetized with isoflurane. The ovaries from 12 HIS/Hiph and 15 HIS/Mz were harvested at 4 weeks of age. We did not examine estrus cycles at the time of sampling because the effect of estrus cycles is minimal as HIS/Hiph has MOFs and DNOs at neonatal age [4]. We collected 287 HIS/Hiph delivery records from October 2015 to January 2021 and 51 HIS/Mz delivery records from December 2019 to December 2020. From these, we analyzed the number of pups, their sex ratio, approximate age of mother, and the correlation between the number of pups and mother ages. Our previous study used neonatal HIS/Hiph rat up to 7 days and 6–8 weeks of age after attaining puberty and experiencing several estrus cycles [3,4]. The appearance of follicular disturbances (DNOs or MOFs) varies with the development of follicular stages, age, puberty, estrous cycles, and other factors. Furthermore, MOFs occur more frequently in the ovaries of fetal and juvenile females than in adults [10]. In this study, we compared ovarian histology at the onset of puberty between the HIS/Hiph and HIS/Mz strains at 4 weeks of age.

Each tissue was fixed in 10% neutral buffered formalin (NBF) for histological analysis, 4% paraformaldehyde (PFA) for immunohistochemistry and terminal deoxynucleotidyl transferase dUTP nick end labeling (TUNEL) assays, or 2.5% glutaraldehyde (GTA) in 0.1 M phosphate buffer (PB) for ultrastructural analysis.

### 2.2. Histological Analysis and Histoplanimetry

Paraffin-embedded ovaries from HIS/Hiph and HIS/Mz (five from each strain) were cut to a thickness of 3 μm to obtain whole ovarian sections. The sections were stained with hematoxylin–eosin (HE) or periodic acid Schiff–hematoxylin (PAS-H) to comparatively examine the ovarian histomorphology between HIS/Hiph and HIS/Mz. The stained sections were scanned using a NanoZoomer 2.0 RS virtual slide scanner (Hamamatsu Photonics, Shizuoka, Japan). Data obtained were used for histoplanimetry. Histological images for microscopic examinations were obtained using a model BZ-X710 microscope (Keyence, Osaka, Japan).

For histoplanimetry analysis using HE-stained sections, ovarian follicles were classified on the basis of their granulosa cell (GC) layer, as PrF (one incomplete or complete layer of flattened squamous epithelium and one layer of a mixture of flattened squamous and cuboidal GCs), primary (PF; one layer of cuboidal GCs), secondary (SF; two to six layers of GCs), and tertiary follicles (TF; more than six layers of GCs), according to previous studies in rabbits [26] and neonatal SH [4]. The number of developing follicles was counted every seventh section and was multiplied by seven to determine the total number of follicles per ovary [4,24,27]. Only follicles containing visible oocytes were counted to avoid double counting. The number and percentage of developing follicles were calculated from the total number of follicles.

Ovarian follicles containing two or more oocytes (defined as MOFs) or oocytes containing two nuclei (defined as DNOs) were classified according to our previous study [3,4] and enumerated. The percentages of MOFs and DNOs in all follicle types were also calculated.

### 2.3. Immunohistochemistry

Immunohistochemistry was performed to evaluate cell proliferation by targeting the marker of proliferation Ki-67 (Ki67) and proliferating cell nuclear antigen (PCNA). Positive oocytes for tumor protein p63, associated with female fertility [28], were also examined. Deparaffinized 3 μm thick sections of the ovaries of HIS/Hiph and HIS/Mz were incubated in 10 mM citrate buffer (pH 6.0) for 15 min at 110 °C. The sections were treated with 0.3% hydrogen peroxide/methanol solution for 20 min to quench the endogenous peroxidase activity. Sections were blocked using 10% normal goat serum (SABPO kit; Nichirei Bioscience, Tokyo, Japan) for Ki67 and p63 or 5% normal donkey serum (Sigma Aldrich, St. Louis, USA) for PCNA. Sections were incubated overnight with rabbit anti-Ki67 (1:800; ab15580; Abcam; Cambridge, UK), goat anti-PCNA (1:3000; sc9857; Santa Cruz Biotechnology, CA, USA), and mouse anti-p63 (1:50; sc8431; Santa Cruz Biotechnology) at 4 °C. The sections were then treated with biotinylated goat anti-rabbit IgG (SABPO kit; Nichirei Bioscience) (for Ki67), donkey anti-goat IgG (1:400; Santa Cruz) (for PCNA), or goat anti-mouse IgG (1:200; Southern Biotech) (for p63) for 30 min at room temperature. This was followed by incubation with streptavidin–horseradish peroxidase using the SABPO kit (Nichirei Bioscience) for 30 min, followed by incubation with 3,3-diaminobenzidine tetrahydrochloride. Finally, the sections were counterstained with hematoxylin and dehydrated using ascending grades of alcohol. The stained sections were examined using a BZ-X710 microscope (Keyence).

### 2.4. TUNEL Assay

Apoptosis was assessed by TUNEL assay using the In Situ Apoptosis Detection Kit (ab206386; Abcam). The HIS/Hiph and HIS/Mz ovaries (three from each strain) were fixed in 4% PFA, embedded in paraffin, and sectioned to a thickness of 3 μm at 30 μm intervals, and the five largest ovarian cross-sections were used. The staining procedure was performed according to the manufacturer’s instructions and a previously published method [4]. A model BZ-X710 microscope (Keyence) was used to obtain the images.

### 2.5. Electron Microscopy

The HIS/Hiph and HIS/Mz ovaries (four from each strain) were collected and immediately fixed with 2.5% GTA in 0.1 M PB for 4 h at 4 °C, followed by post-fixation with 1% osmium tetroxide (OsO_4_) in 0.1 M PB for 2 h. The specimens were dehydrated with ascending grades of alcohol and embedded in epoxy resin (Quetol 812 Mixture; Nisshin EM, Tokyo, Japan). The epoxy blocks were cut to 60 nm thickness. Ultrathin sections (at least 10 sections from each ovary) were mounted on grids and stained with uranyl acetate and lead citrate for 15 min and 10 min, respectively. Transmission electron microscopy (TEM) of the stained sections was performed using a model JEM-1210 transmission electron microscope (JEOL, Tokyo, Japan).

### 2.6. Statistical Analysis

The results are expressed as the mean ± standard error (SE) and were analyzed using nonparametric methods. Follicles at each developmental stage and the MOFs and DNOs of the two strains were compared using the Mann–Whitney U-test (*p* < 0.05). The Kruskal–Wallis test was used to compare all types of developing follicles and their MOFs and DNOs within the same strain, and multiple comparisons were performed using Scheffé’s method (*p <* 0.05).

## 3. Results

### 3.1. Comparative Results of Recorded Data of HIS/Hiph and HIS/Mz

Table 1 shows numbers of HIS/Hiph and HIS/Mz pups produced, averaging five to six pups per litter, which was within normal limits. Briefly, 287 HIS/Hiph deliveries were recorded between October 2015 and January 2021, when the highest number of pups per individual was 13 while the lowest was one, and 51 HIS/Mz deliveries were recorded between December 2019 and December 2020, when the highest number of pups per individual was 10 while the lowest was 1. The mother age and sex ratio and were almost the same.

Figure 1 shows the mother age in days of delivery and its relationship with the number of pups according to recorded history. There was no correlation between both parameters in both strains. Although the purpose of breeding and recording was different, they did not measure the reproductive ability. In CRs, the average number of pups was 5.0 at the first birth, 5.9 at the second birth, 6.6 at third 3rd and fourth births, and 6.3 at the fifth birth [29]. The results approximately match the number of weeks to the first birth and the description in the review.

### 3.2. Comparative Ovarian Histology of HIS/Hiph and HIS/Mz

Figure 2 shows the ovarian histology of HIS/Hiph and HIS/Mz. Early developing follicles, such as PrF and PF, were found in the cortex of both strains. Furthermore, the follicles that progressed to later developing stages, such as SF and TF, were localized in the superficial or deep regions of the cortex. In the medulla, the ductal structures resembling remnants of rete ovarii [30] were frequently found in HIS/Hiph (Figure 2a), whereas more numerous interstitial cells (ICs) were found in the medullary stroma of HIS/Mz (Figure 2b) than in HIS/Hiph.

For histoplanimetry, we quantified the number and percentage of each developing follicle in HIS/Hiph and HIS/Mz. In both strains, the total number and percentage of developing follicles significantly decreased with the progression of folliculogenesis (Figure 2c,d). HIS/Hiph revealed nearly twofold higher numbers of PrF, PF, and SF than those of HIS/Mz, with significant differences, while TF was comparable in both strains (Figure 2c). Conversely, for the number of follicles in each follicle stage to total number of follicles (indicated as a percentage), a significant difference was observed only in PF between HIS/Hiph and HIS/Mz (Figure 2d).

### 3.3. MOFs Found in the Ovaries of HIS/Hiph and HIS/Mz

Figure 3 shows the appearance of the MOFs. The ovaries of the HIS/Hiph strain possessed MOFs in the PrF, PF, SF, and TF stages (Figure 3a–d). The oocytes contained in the same MOFs were closely arranged, but each oocyte was surrounded by the zona pellucida (ZP) (Figure 3a–c) or clearly separated by GCs (Figure 3d). In the ovaries of HIS/Mz, MOFs were found up to SF, but not in TF (Figure 3e–h). Most oocytes were closely arranged in the same MOF (Figure 3e–g), but rarely separated by GCs. The oocytes of MOFs separated by either ZP or GCs were more frequently observed in HIS/Hiph than in HIS/Mz. We previously reported that the ovaries of mice, including C57BL/6N and ICR, have no MOFs [4].

Histoplanimetry analysis revealed that HIS/Hiph had 2–6-fold higher numbers of MOFs showing the features of PrF, PF, and SF than HIS/Mz (*p* < 0.01, Figure 3i). Within the HIS/Hiph strain, the number of PF-type MOFs was significantly higher than that of other types of MOFs *(p* < 0.01), whereas PrF- or PF-type MOFs were significantly higher in number than the SF-type MOFs in the HIS/Mz strain (Figure 3i). The ratio of the number of PF- or SF-type MOFs to the total number of follicles was significantly higher in HIS/Hiph than in HIS/Mz (*p* < 0.01), although PrF-type MOFs showed no strain difference (Figure 3j). Moreover, in HIS/Hiph, the number of SF-type MOFs was significantly higher than that of PrF- or TF-type MOFs (*p* < 0.01), and the number of PF-type MOFs was significantly higher than that of PrF-type MOFs (*p* < 0.05). No significant differences were observed among the follicle stages of the MOFs in the HIS/Mz (Figure 3j).

Table 2 summarizes the number of oocytes observed in the single MOF at each follicular development stage in the ovaries of HIS/Hiph and HIS/Mz. In both strains, most MOFs had two oocytes in a single follicle, and the appearance frequency of MOFs containing two oocytes increased with the progression of folliculogenesis, but TF in HIS/Mz did not contain multiple oocytes. The number of oocytes inside the MOFs varies among species: 2–24 oocytes in rabbits [26], 2–9 in goats [31], 2–17 in dogs [32], and 2–9 in cattle [6]. Interestingly, HIS/Hiph tended to have more oocytes in a single MOF than in HIS/Mz. Although four or more oocytes were only found in PrF-type MOFs in HIS/Mz, they were observed up to the TF stage in HIS/Hiph.

### 3.4. Unique Morphologies Found in the MOF of HIS/Hiph and HIS/Mz

Although the MOFs were mostly round to oval in shape (Figure 3a–h), two types of characteristic structures were also observed (Figure 4). First, two individual follicles were partially fused by sharing GCs, and the two follicles were surrounded by the same basal lamina (BL) (Figure 4a–c). The connecting region of each follicle was mildly constricted and was found in the later stages of developing follicles, including SF and TF. This type of MOF was observed in HIS/Hiph (Figure 4a–c), whereas it was partial in the HIS/Mz MOFs (Figure 4d–f), as observed in serially sectioned ovaries.

In the other structure, two individual follicles were connected by a narrow bridge of GCs (Figure 4g). These MOFs showed more irregular shapes, and the connecting region of each follicle was severely constricted compared to BL constricted-type MOFs, as described above. Interestingly, these GC bridge-type MOFs were frequently observed in the SF and TF stages of MOFs. These GC bridge-type MOFs were frequently observed only in HIS/Hiph and not in HIS/Mz.

### 3.5. DNOs Found in the Ovaries of HIS/Hiph and HIS/Mz

DNOs, which are defined as oocytes of SOF containing two nuclei, were found in HIS/Hiph but not in HIS/Mz (Figure 5a–d). Previously, we reported that DNOs were found up to SF in HIS/Hiph (0–7 days after birth, 6–8 weeks old SH) [3,4]. In 4-week-old HIS/Hiph, DNOs were found in all types of developing follicles (Figure 5a–d). The nuclei of DNOs were closely localized to each other in the oocytes within the early developing follicles, including PrF and PF (Figure 5a,b), but were separated in SF and TF (Figure 5c,d). Most DNOs in the PrF and PF were localized in the peripheral region of the cortex, and those of later stages were found in the superficial and deep cortex areas, with single-nucleated oocytes. These DNOs were not found in the ovaries of C57BL/6N and ICR mice [4].

Next, we enumerated the number and percentage of DNOs. In HIS/Hiph, PrF showed the highest number of DNOs, whereas the number of DNOs significantly decreased in the consecutive follicles (*p* < 0.01; Figure 5e). Furthermore, the percentage of DNOs in each follicular stage appeared to be higher in TF than in other types of follicles, without a significant difference (*p* = 0.167). DNOs were not detected in HIS/Mz.

### 3.6. Follicular Cell-Cycle Arrest and Oocyte Death in the Ovaries of HIS/Hiph and HIS/Mz

We then evaluated oocyte death as a factor affecting the severe difference in the total number of each follicle type and the appearance of MOFs and DNOs in both strains. First, we examined the expression of p63 in oocytes, which promotes cell-cycle arrest and inhibits apoptosis of oocytes [33], in the ovaries of HIS/Hiph (Figure 6a,b) and HIS/Mz (Figure 6c). Oocytes of the early developing follicles, including PrF, PF, and SF, with two layers of GCs, expressed p63 in both strains (Figure 6a,c). Oocytes in the MOFs showed the same positivity for p63 as the SOFs in both strains (Figure 6a,c). In the HIS/Hiph strain, the nuclei of DNOs in the SOF of PrF, PF, and two layers of GCs were also positive for p63 (Figure 6b). Consistent with this, we did not observe any apoptotic oocytes in SOF, MOF, or DNOs, but luteinized GCs showed positive reactions in both strains by TUNEL assay (Figure 6d).

### 3.7. Proliferative Activity of the Oocytes in the Ovaries of HIS/Hiph and HIS/Mz

We also evaluated the proliferative activities in the oocytes of SOF, MOF, and DNOs (Figure 7). In mammalian ovaries, although immunoreactivity for PCNA, a cell proliferation marker, was detected in the oocytes of ruminants such as cattle [12] and sheep [34], it was observed in the GCs and theca cells of mice [35]. In HIS/Hiph, oocytes in SOFs and MOFs or DNOs also showed positivity for PCNA regardless of follicular developmental stage (Figure 7a–c and Appendix A). The positivity for PCNA was clearer in the nuclei of oocytes of SOFs or MOFs and of DNOs in HIS/Hiph (Figure 7a–c) than in those of SOFs and MOFs in HIS/Mz (Figure 7d,e).

Conversely, Ki67, another cell proliferation marker, was only observed in the nuclei of GCs, but not in those of oocytes in the HIS/Hiph strain, as shown in TF-type SOFs and SF-type MOFs (Figure 7f,g). No positive reactions were observed in the DNOs of HIS/Hiph (Figure 7h). In HIS/Mz, Ki67-positive reactions were not observed in the nuclei of oocytes, and those of GCs were faint in SOFs and MOFs (Figure 7i,j).

### 3.8. Ultrastructures of SOFs, MOFs, and DNOs in the Ovaries of HIS/Hiph and HIS/Mz

Figure 8 shows the results of TEM analysis for SOFs, MOFs, and DNOs in the ovaries of HIS/Hiph and HIS/Mz at 4 weeks of age. In HIS/Hiph, SOFs containing single-nucleated oocytes (SNOs), determined as PrF, contained abundant mitochondrial clouds (MCs) dispersed throughout the cytoplasm, and MCs showed grape-like structures (Figure 8a). In MOF, MCs were also abundant in each oocyte, and these oocytes were clearly distinguished by ZP (Figure 8b). In DNOs, MCs were observed between double nuclei (Figure 8c). In SOFs of HIS/Mz classified as PF, SNOs were surrounded by cuboidal GCs, and they had a few MCs with lower electron density than those in HIS/Hiph (Figure 8d). MOF of HIS/Mz also showed fewer numbers and lower electron density of MCs than HIS/Hiph (Figure 8e). No DNOs were observed in HIS/Mz, similar to the histology results.

## 4. Discussion

This study demonstrated the strain-specific histological differences in the ovaries of two inbred strains of SH. HIS/Hiph had MOFs and DNOs in all stages of developing follicles, but HIS/Mz had fewer MOFs and no DNOs (Figure 3 and Figure 5). Moreover, the estimated follicle population of the HIS/Mz strain was almost half that of the HIS/Hiph strain (Figure 2). Higher numbers of MCs were also observed in the HIS/Hiph strain than in HIS/Mz (Figure 8). These results indicate that ovarian morphology differed between SH strains, and that HIS/Hiph clearly showed unique ovarian phenotypes, such as numerous MOFs and the presence of DNOs.

During folliculogenesis, follicular morphology changes as the oocyte grows and the surrounding cells proliferate. The major morphological distinction between domestic animals and rodents is the sheer abundance of PrFs in the latter, located throughout the ovarian cortex. The ovaries of laboratory mouse strains contain follicles of varying stages of development in the order of 3000 to 5000 [36]. This number declines over time as most follicles die via atresia, with a small number surviving to ovulation in each estrus cycle [37]. In fact, ICs, derived from the internal theca of atretic follicles in rodents [38,39], were more abundant in HIS/Mz than in HIS/Hiph. We considered that this result might reflect the difference in folliculogenesis between SH strains, including follicle atresia and oocyte death. However, there were no clear strain-specific differences in the positivity of oocytes for p63 (Figure 5a–c), a regulator of the cell cycle in mouse germ cells [33], as well as oocyte death examined by TUNEL assay (Figure 6d) at 4 weeks of age. Therefore, we considered that the differences in the number of developing follicles would be formed during the embryonic or neonatal period, but not at 4 weeks of age.

Furthermore, litters averaged five to six pups in CRs [17], and our results indicated that the number of pups was within the normal range. In the present study, there was no age-related decline in pup numbers in both strains. Although the history of breeding for HIS/Hiph (5 years) and HIS/Mz (1 year) was recorded for the maintenance of each strain, it did not include an evaluation of reproductive ability. To examine reproductive ability, we have to compare the number of pups in the first delivery from the mating of sexually mature males and immature females. Table 1 did not take this point into account; therefore, the number of deliveries might have affected this result. Therefore, these points are considered to be limitations of our study. This information would be helpful for future studies to determine the fertility status and its age-related alternations in HIS/Hiph and and HIS/Mz.

The frequency of MOFs is important for further studies to evaluate the mechanisms and fertility status of such disturbances in mammals. However, the frequency of MOFs in the ovaries is species- and individual-dependent. In opossum (<3 months), dogs (<7 months), and pigs (<5 months), MOFs are more common in the ovaries of younger than older individuals [7,10,40]. For dogs, there is disagreement on this topic. Some authors have shown that MOFs are less common in older dogs than at the beginning of their reproductive lives [7,13], but other authors have reported no difference in the numbers of MOFs between young and adult females [32]. For immature hamster ovary, not only is a large number of polyovular follicles but a quantitative fluctuation in the number of such follicles is observed in the animals of different ages [41]. The formation of these disturbances in mammals occurs owing to the failure of germ cell division during the early stages of folliculogenesis, such as in humans [42] or bovine [2]. Recently, a new theory for MOF formation suggested that follicles are fused by the invasion of GCs toward other follicles [28]. In SH, the strain difference in the frequency of MOFs was first demonstrated in this study. Regarding breed, these structures were observed more frequently in mongrels (52.3%) than in purebred animals (25.5%) [7]. Notably, the number and percentage of PF-type MOFs were higher than those of PrF only in HIS/Hiph (Figure 3i,j), indicating that MOFs can develop after the formation of follicles. Furthermore, BL constricted-type and GC bridge-type MOFs were also found in SH; in particular, these were clearer in HIS/Hiph than in HIS/Mz. We considered that BL constricted-follicle structures indicated “BL breaching (BLB)”, which is one of the most important characteristics for the formation of MOFs. BLB might be a result of (i) decreased synthesis of extracellular matrix proteins that comprise the BL, and (ii) increased degradation of BL more than that required for normal remodeling during follicular growth [43]. Furthermore, GC bridge structure resembling “granulosa cell invasiveness (GCI)” has been reported in inbred rat [28], and the aberrant initiation of cellular invasion by the GCs results in the eventual breaching of the follicle wall allowing GCs to invade the perifollicular stroma or joining of adjacent follicles to form MOFs. In fact, the expression of proliferative cell markers PCNA and Ki67 was detected in GCs of both strains, and Ki67 positivity was stronger in HIS/Hiph than in HIS/Mz (Figure 7). These results indicate that fusion of two or more SOFs, or BLB or GCI, occurs in SH, especially in HIS/Hiph, showing higher proliferating activities of GCs to form MOFs. Furthermore, in the giant panda, the existence of twin oocytes in the follicle increases the rate of twinning [44] and, recently, in humans, successful live births from conjoined oocytes have been reported [45]. Therefore, the difference between strains in abundance of MOFs and DNOs may play a role in twin research.

During comparative histological analysis of SH strains, we found another important strain difference that DNOs, found in all types of follicles in HIS/Hiph ovaries, were lacking in HIS/Mz. Double-nucleated cells can also be observed in other cell types, such as hepatocytes or trophoblasts, and these double nuclei indicate intensive transcriptional activity [46]. There are no reports of DNOs in other mammals, except that binuclear oocytes are usually found in opossums [40]. Multinucleated oocytes have also been described during the initial stages of oogenesis in some amphibian species, such as *Ascaphus truei*, in which oogenesis involves eight nuclei [47], whereas, in *Leiopelma hochstetteri,* oocytes with two nuclei have been described [48]. Moreover, the number and percentage of DNOs at 1 month of age were lower than those of neonates [4] and higher than those of 6–8 weeks of age [3]. Therefore, the DNOs might decrease with the development of ovaries. Unlike MOFs, the number of DNOs was the highest in PrF and declined with follicle development. In addition, oocytes were Ki67-negative/PCNA-positive in the examined SH, and PCNA positivity was clearer in HIS/Hiph than in HIS/Mz. These results indicated that PCNA-positive oocytes did not reflect proliferative cells, but the other cell types. Although PCNA is a general proliferation marker, PCNA expression in oocytes is associated with apoptosis rather than proliferation [35,49]. Thus, strong expression of PCNA in HIS/Hiph ovaries (Figure 7a) alleviates apoptosis of oocytes and increases PrF assembly and formation of follicles with DNOs, as compared to HIS/Mz ovaries showing weak expression.

The presence of MCs in the SOFs, MOFs, and DNOs was clarified by TEM analysis of ovaries of HIS/Hiph and HIS/Mz strains (Figure 8). Higher numbers of MCs were observed in the HIS/Hiph strain oriented throughout the cytoplasm as a grape-like structure than in the HIS/Mz strain. The presence of MCs in oocytes has been termed Balbiani bodies (Bbs), which contain a large RNA-protein granule, and they are universally conserved in the oocytes of insects [50], fish [51], rodents [52], and humans [53]. It is still not known how the structural and molecular components of Bbs are assembled into such morphologically intricate complexes. The most important function of mitochondria is to supply cells with metabolic energy (ATP) generated by oxidative phosphorylation [54]. Therefore, we hypothesize that the presence of more MCs might have contributed to more oocytes in the HIS/Hiph strain than in the HIS/Mz strain by maintaining oocyte reserves.

## 5. Conclusions

In conclusion, our findings clarify the comparative histology of SH strains, especially focusing on the quantity and quality of normal oocytes with MOFs and DNOs. The two inbred strains with different phenotypes facilitate the analyses of the genetic factors that determine MOFs and DNOs. Our results would be helpful in understanding folliculogenesis, molecular mechanisms, and fertility status, as well as their disturbances, including the appearance of MOFs and DNOs in several mammalian species, including humans.

## Figures and Tables

**Figure 1 animals-11-01768-f001:**
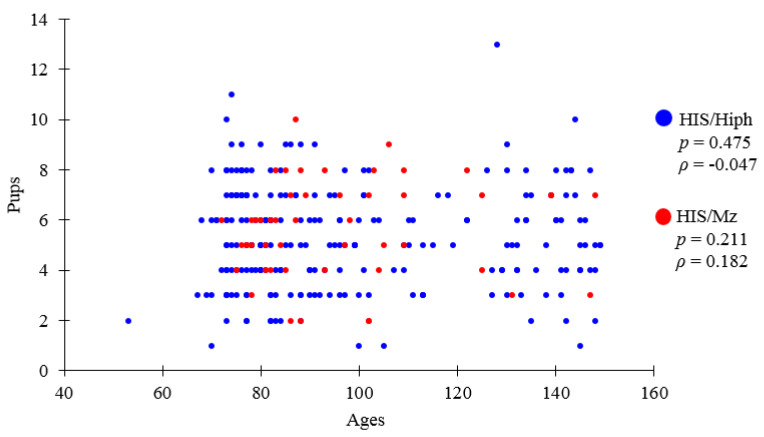
Numbers of pups and mother age in days at delivery. The total number of pups and mother age in days at delivery in the HIS/Hiph and HIS/Mz strains. Values are the mean ± standard error. Correlation between the number of pups and mother age in days at delivery in the HIS/Hiph and HIS/Mz strains. Spearman’s rank correlation coefficient (*n* = 234 deliveries in HIS/Hiph and 49 deliveries in HIS/Mz).

**Figure 2 animals-11-01768-f002:**
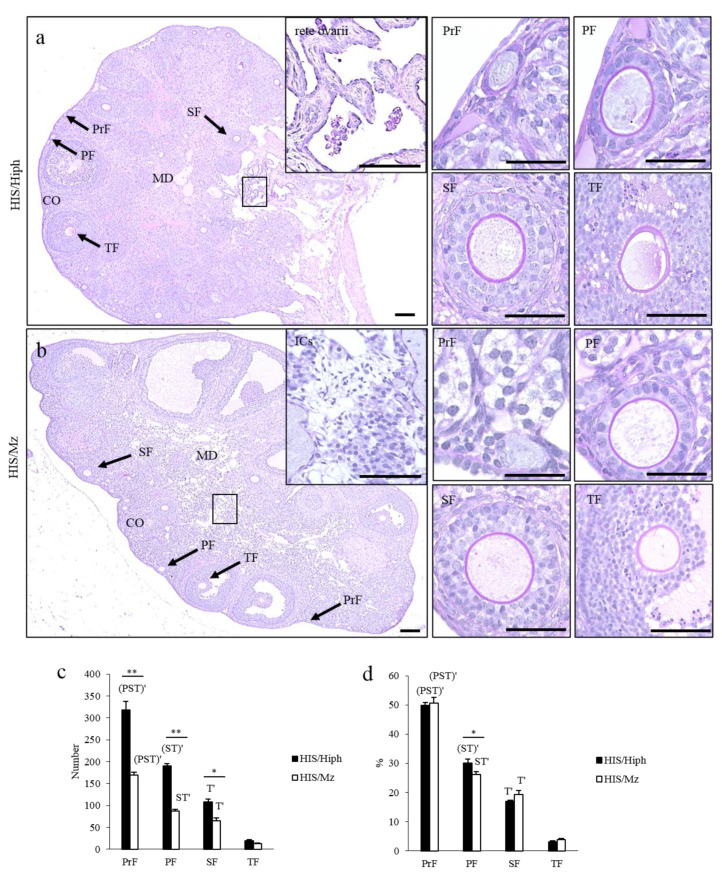
Histology of the ovaries of HIS/Hiph and HIS/Mz. (**a**,**b**) Histological observation of periodic acid Schiff–hematoxylin-stained ovary section from 4- week- old HIS/Hiph and HIS/Mz. The classified follicles shown in panels a and b (arrows) are further magnified to highlight primordial (PrF), primary (PF), secondary (SF), and tertiary (TF) follicles in both strains. Panels (**a**) and (**b**) show the regions of the ovary divided as cortex (CO) and medulla (MD). Panel a also shows the remnants of the rete ovarii (inset). Panel b shows the interstitial cells (ICs) (inset). The square area indicates the same area of insets. Scale bars = 200 µm or 50 µm (magnified and inset). (**c**,**d**) The total number and percentage of each follicle type classified as PrF, PF, SF, and TF in the ovary of HIS/Hiph and HIS/Mz. The percentage was calculated from the number of each follicle type/total number of follicles. Values are the mean ± standard error. Asterisks indicate significant differences between HIS/Hiph and HIS/Mz within a follicle type (Mann–Whitney *U*-test, * *p <* 0.05 and ** *p <* 0.01), and letters (Pr, P, S, and T) denote significant differences in the number of PrF, PF, SF, and TF in same strain (Kruskal–Wallis test followed by Scheffe’s method, *p* < 0.05). Dashes beside letters indicates a highly significant difference (*p* < 0.01). *N* = 5 in each strain.

**Figure 3 animals-11-01768-f003:**
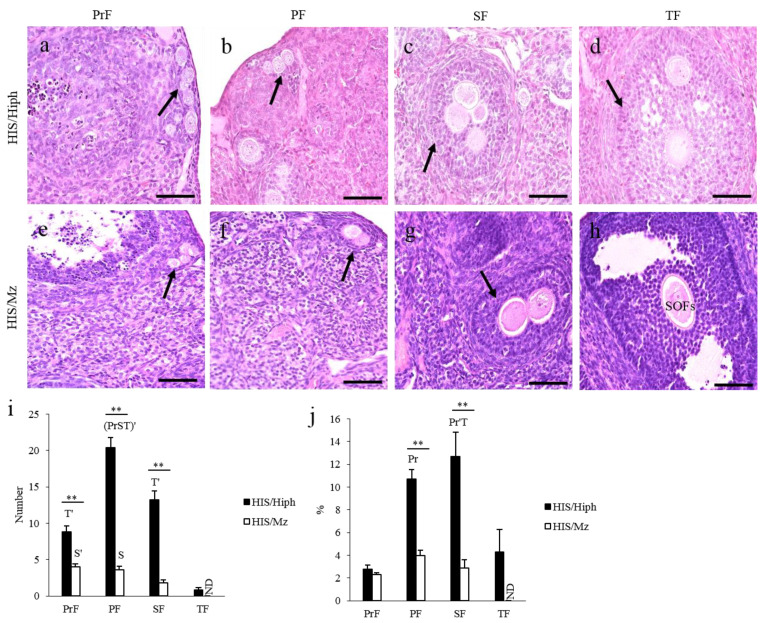
Multi-oocyte follicles (MOFs) in the ovaries of HIS/Hiph and HIS/Mz. (**a**–**h**) Histological observation of MOFs from hematoxylin–eosin-stained sections of the ovaries of 4-week-old HIS/Hiph (**a**–**d**) and HIS/Mz (**e**–**g**). MOFs denoted by the arrows are observed in follicle types classified as primordial follicle (PrF), primary follicle (PF), secondary follicle (SF), and tertiary (TF) follicle in HIS/Hiph (**a**–**d**) and PrF, PF, and SF in HIS/Mz (**e**–**g**). Panel h shows the TF-type single-oocyte follicle (SOF). Scale bars = 100 µm. (**i**,**j**) The total number and percentage of MOFs (follicular type MOFs/number of follicles in each type) classified as each follicle type in the ovary of HIS/Hiph and HIS/Mz. Values are the mean ± standard error. Asterisks indicate significant differences within follicle types between HIS/Hiph and HIS/Mz strains (Mann–Whitney *U*-test, ** *p* < 0.01). Pr, P, S, and T denote significant difference in the number of PrF, PF, SF, and TF, respectively, in HIS/Hiph and HIS/Mz (Kruskal–Wallis test followed by the Scheffé’s method, *p* < 0.05). Dashes beside letters indicate a highly significant difference (*p* < 0.01). *N* = 5 in each strain. ND: Not detected.

**Figure 4 animals-11-01768-f004:**
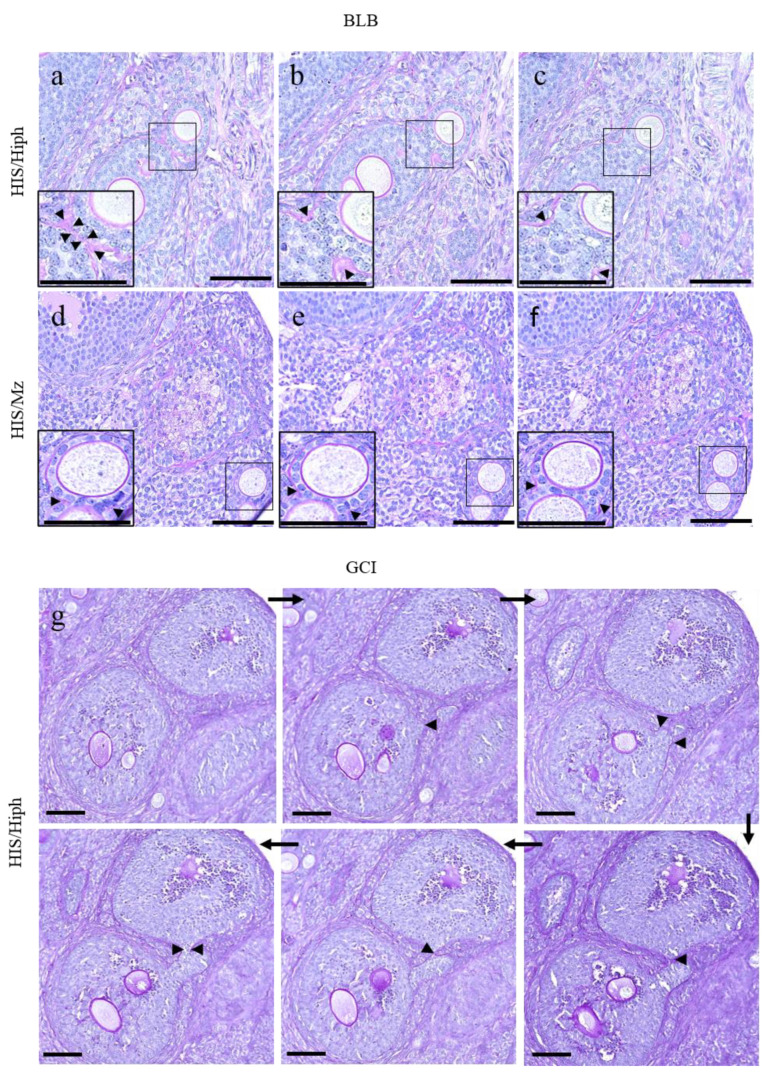
Unique morphological characteristics of the multi-oocyte follicles (MOFs) in the ovaries of HIS/Hiph and HIS/Mz. (**a**–**f**) Histological observation of basal lamina (BL) constricted-type MOFs in periodic acid Schiff (PAS)–hematoxylin-stained serial sections from the ovaries of 4-week-old HIS/Hiph (**a**–**c**) and HIS/Mz (**d**–**f**). BL constricted-type MOFs are clearly observed in HIS/Hiph (**a**–**c**), but they are not clear in HIS/Mz (**d**–**f**). (**g**) Granulosa cell (GC) bridge-type MOFs in PAS-stained serial sections in the ovary of HIS/Hiph. Arrows indicate the order to next serial sections. Arrowhead shows GC bridge regions. Scale bars = 100 µm or 50 µm (inset).

**Figure 5 animals-11-01768-f005:**
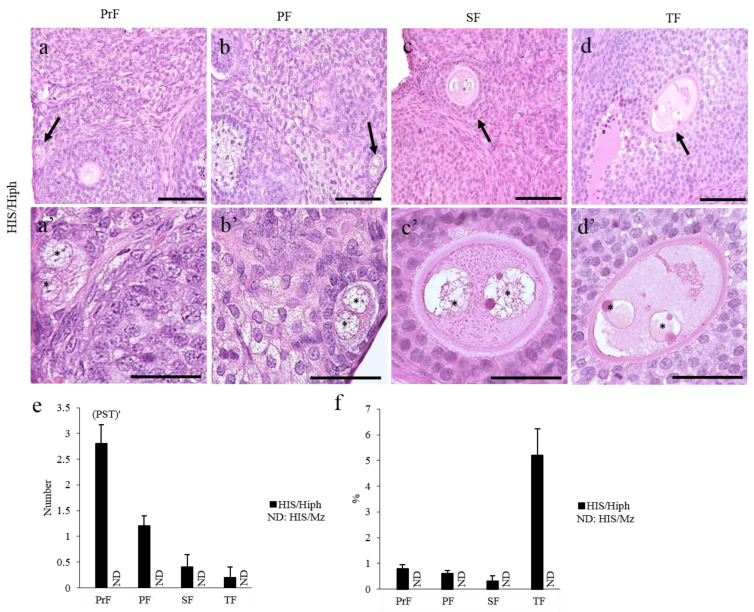
Double-nucleated oocytes (DNOs) in the ovaries of HIS/Hiph and HIS/Mz. (**a**–**d** and **a’**–**d’**) Histological observation of DNOs from hematoxylin–eosin-stained sections of the ovaries of 4-week-old HIS/Hiph and HIS/Mz. Two nuclei are observed within a single oocyte classified as primordial follicle (PrF), primary follicle (PF), secondary follicle (SF), and tertiary follicle (TF) in panels (**a**–**d**) (arrows) and (**a’**–**d’**) (magnified), respectively, but DNOs are absent in HIS/Mz. Each nucleus (asterisk) shares the same cytoplasm. Scale bars = 100 µm or 50 µm (magnified). (**e**,**f**) The total number and percentage of DNOs classified as developing follicles in the ovary of HIS/Hiph. Values are presented as the mean ± standard error. Pr, P, S, and T denote significant differences in the number of PrF, PF, SF, and TF, respectively, in HIS/Hiph (Kruskal–Wallis test followed by the Scheffé’s method, *p* < 0.05). Dashes beside letters indicate a highly significant difference (*p* < 0.01). *N* = 5 in each age. ND: Not detected.

**Figure 6 animals-11-01768-f006:**
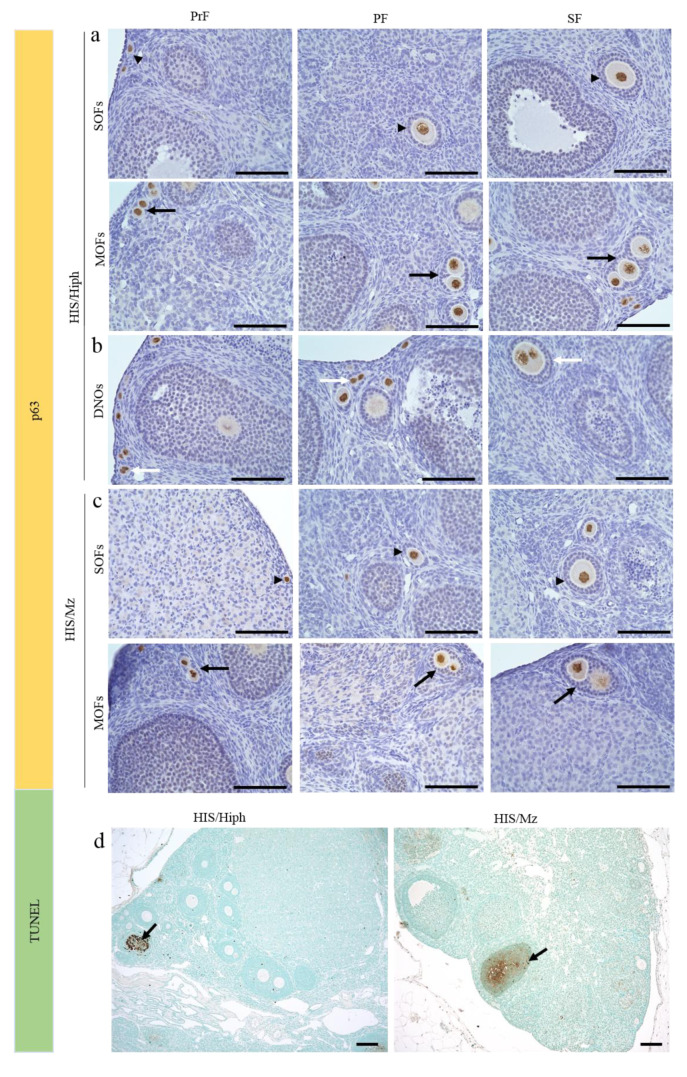
Evaluation of oocyte death in the ovaries of HIS/Hiph and HIS/Mz. (**a**–**c**) Immunohistochemistry for p63. Single-oocyte follicles (SOFs) classified as primordial (PrF), primary (PF), and secondary (SF) follicles (arrowheads), as well as multi-oocyte follicles (MOFs) (arrows) and double-nucleated oocytes (DNOs) (white arrows, found only in HIS/Hiph), containing p63-positive oocytes in 4-week-old HIS/Hiph (**a**,**b**) and HIS/Mz (**c**). (**d**) TUNEL assay. Panel shows lack of apoptotic oocytes, but granulosa cells (arrows) in the luteinized follicles show positive reactions in both strains. Scale bars = 100 µm (**a**–**c**) or 200 µm (**d**).

**Figure 7 animals-11-01768-f007:**
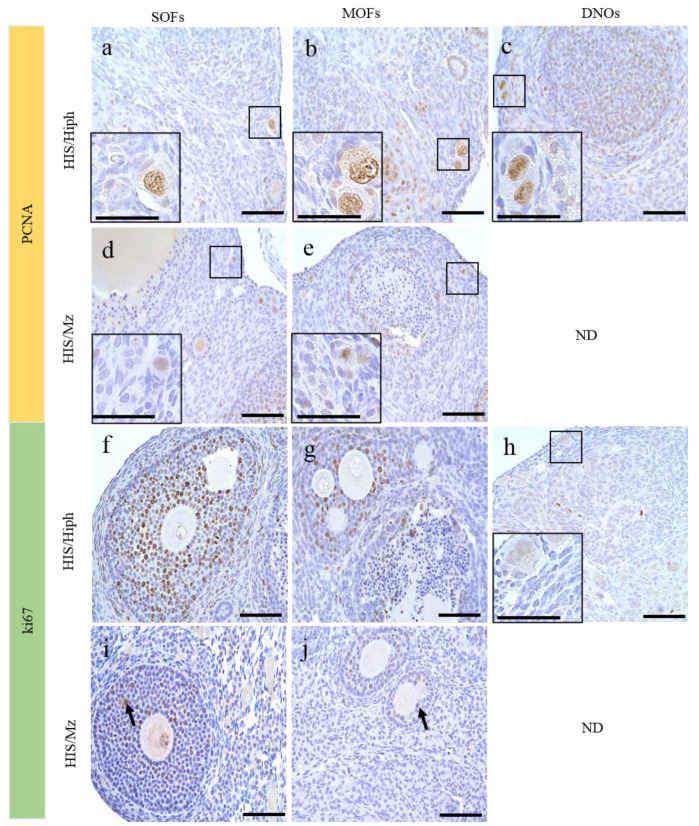
Evaluation of proliferative activity of oocytes in HIS/Hiph and HIS/Mz. (**a**–**e**) Immunohistochemistry for PCNA. PCNA-positive oocytes in single-oocyte follicles (SOF, **a**), multi-oocyte follicles (MOFs, **b**), and double-nucleated oocytes (DNOs, **c**) are observed in the primordial follicles (PrFs) in HIS/Hiph (**a**–**c**). PCNA-positive SOFs and MOFs (weak expression) are observed in PrFs in HIS/Mz (**d**,**e**). (**f**–**j**) Immunohistochemistry for Ki67. Ki67 positivity is are clearly observed in the granulosa cells (GCs) in tertiary follicle-type SOF (**f**) and secondary follicle type-MOFs (**g**) in HIS/Hiph. No positive reactions are observed in oocytes (**f**,**g**), GCs in primordial follicles, and DNOs (**h**) in HIS/Hiph. A few Ki-67-positive GCs are observed in the SOFs and MOFs (arrows) of HIS/Mz. Scale bars 100 µm or 50 µm (inset). ND: Not detected.

**Figure 8 animals-11-01768-f008:**
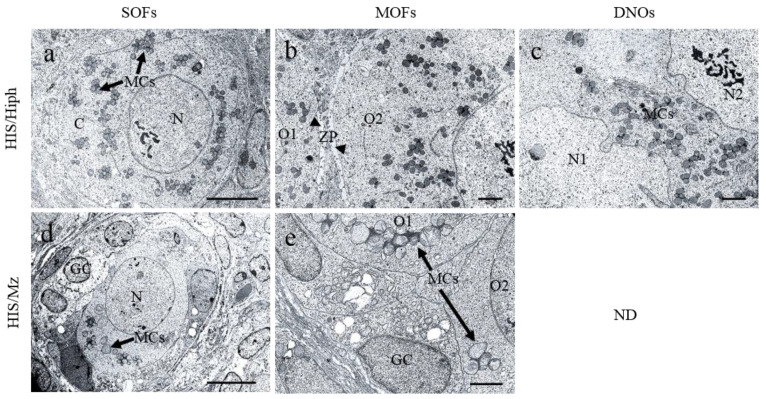
Ultrastructures of ovarian follicles and oocytes in HIS/Hiph and HIS/Mz. Panels are transmission electron microscopy images of the ovary sections of 4-week-old HIS/Hiph and HIS/Mz. Panels (**a**–**c**) and (**d**,**e**) show representative images of HIS/Hiph and HIS/Mz ovaries, respectively. Panel (**a**) indicates a single-oocyte follicle (SOF) with a clear round-shaped nucleus (N) and mitochondrial clouds (arrows) distributed throughout the cytoplasm and oriented as a grape-like structure. Panel (**b**) shows a multi-oocyte follicle (MOF) with differentiated oocytes (O1 and O2) separated by the clear zona pellucida (ZP) (arrowhead). Panel **c** shows double-nucleated oocytes (DNOs) and nuclei (N1 and N2) that are separated by mitochondrial clouds (MCs) (arrows). Panels (**d**,**e**) show the SOF and MOF in the HIS/Mz ovary; a few MCs (arrows) are observed in SOFs and MOFs, and each MC shows a low electron density as compared to that of HIS/Hiph MCs. Scale bars = 5 µm in panel (**a**); 2 µm in panels (**b**,**c**,**e**); 10 µm in panel d. ND: Not detected.

**Table 1 animals-11-01768-t001:** The number of pups recorded in the inbred SH strains, HIS/Hiph and HIS/Mz.

SH Strain	Recorded Period	Mother Age (Days)	Number of Pups		
			Total	Male	Female
HIS/Hiph	2015–2021	97.8 ± 1.6	5.2 ± 0.1 (1–13)	2.8 ± 0.1 (0–8)	2.4 ± 0.2 (0–8)
HIS/Mz	2019–2020	96.6 ± 2.7	5.7 ± 0.3 (1–10)	3.0 ± 0.2 (1–7)	2.7 ± 0.2 (0–6)

Values are the mean ± SE. *n* = 287 deliveries in HIS/Hiph and 51 deliveries in HIS/Mz. Parentheses show minimum to maximum values.

**Table 2 animals-11-01768-t002:** The number of oocytes observed in the multi-oocyte follicles of each follicular developmental stage in the ovaries of HIS/Hiph and HIS/Mz.

Follicle Type	Multi-Oocyte Follicles					
	Two Oocytes (%)		Three Oocytes (%)		>Four Oocytes (%)	
	HIS/Hiph	HIS/Mz	HIS/Hiph	HIS/Mz	HIS/Hiph	HIS/Mz
Primordial	52.7	82.0	43.3	16.0	4.0	2.0
Primary	62.5	94.5	35.0	5.5	2.5	0.0
Secondary	68.0	100.0	27.5	0.0	4.5	0.0
Tertiary	82.5	0.0	15.8	0.0	1.7	0.0

The results are based on the percentages of multi-oocyte follicles of each follicular type in the ovaries of HIS/Hiph and HIS/Mz.

## Data Availability

The data that support the findings of this study are available from the corresponding author upon reasonable request.

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
