# Peer review of "Comparison of Ovarian Morphology and Follicular Disturbances between Two Inbred Strains of Cotton Rats (*Sigmodon hispidus*)"

_animals, 2021, doi:10.3390/ani11061768_

Round 1
Reviewer 1 Report
I reviewed the manuscript changes and the authors did address several areas of concerned improve the descriptive study. One note is on line 443 where the authors say they used 2 years of breeding data, however table 1 only shows 1 year shows from Dec 2019-2020.
Reviewer 2 Report
This manuscript has been improved, but still inadequate to be published in this journal with the present form. For instance, the word number of the simple summary is almost the same as that of the abstract (~200 words). This reviewer found it no reason to be "simple". It is, preferably, about 1/3-1/2 of the abstract in terms of its word number, but surely depending on the journal policy.
Also, the description in Lines 26-28 is not necessary and could be misleading due to no mention about known MOF in one of the earliest species such as rabbits and hamsters decades ago.
The earliest papers in rabbits and hamsters should be credited in this study.
The reason to use these to inbred strains can be further stressed in terms of, e.g., its significance and/or its general phenomenon seen in other species, or something else.
Reviewer 3 Report
The authors have accomplished my requests and revised the manuscript in a satisfactory manner.
Author Response
Thank you.